# Maternal Baseline Risk Factors for Abnormal Vaginal Colonisation among High-Risk Pregnant Women and the Association with Adverse Pregnancy Outcomes: A Retrospective Cohort Study

**DOI:** 10.3390/jcm12010040

**Published:** 2022-12-21

**Authors:** Junesoo Jeon, Yun-sun Choi, Yejin Kim, Siryeon Hong, Ji-Hee Sung, Suk-Joo Choi, Soo-young Oh, Cheong-Rae Roh

**Affiliations:** 1School of Medicine, Sungkyunkwan University, Seoul 06355, Republic of Korea; 2Department of Obstetrics and Gynecology, Samsung Medical Centre, School of Medicine, Sungkyunkwan University, Seoul 06351, Republic of Korea

**Keywords:** high-risk pregnancy, abnormal vaginal colonisation, maternal risk factor, preterm birth, adverse pregnancy outcome, Gram-negative bacteria, Gram-positive bacteria, *Ureaplasma urealyticum*

## Abstract

Abnormal vaginal colonisation can lead to adverse pregnancy outcomes such as preterm birth through intra-amniotic inflammation. Despite the concern, little is known about its risk factors and impact in pregnant women at high-risk for spontaneous preterm birth. Thus, we conducted this single-centre retrospective cohort study including 1381 consecutive women who were admitted to the high-risk pregnancy unit. The results of vaginal culture at admission were categorised according to the colonising organism: bacteria (Gram-negative or -positive) and genital mycoplasmas. Maternal baseline socioeconomic, and clinical characteristics, as well as pregnancy, delivery, and neonatal outcomes were compared according to the category. Maternal risk factors for Gram-negative colonisation included advanced maternal age, increased pre-pregnancy BMI, a greater number of past spontaneous abortions, earlier gestational age at admission, and IVF. Gram-positive colonisation was likewise associated with earlier gestational age at admission. Genital mycoplasmal colonisation was associated with a greater number of past induced abortions, a lower level of education completed, and a lower rate of multifetal pregnancy and IVF. The neonates from mothers with Gram-negative colonisation had a greater risk of NICU admission, proven early onset neonatal sepsis, and mortality. However, not Gram-positive bacteria or genital mycoplasma was directly associated with adverse pregnancy outcomes.

## 1. Introduction

About 40–50% of preterm births are associated with intrauterine infection [1], most of which occur through ascending vaginal infection [2]. It is well recognised that abnormal vaginal flora (microbiota/microbiome), characterised by a decrease in lactobacilli and overgrowth of aerobic or anaerobic microorganisms, is a contributing factor for ascending infection [3]. Previous studies have indicated that the presence of abnormal vaginal flora was associated with spontaneous preterm birth caused by preterm labour (PTL), preterm premature rupture of membranes (pPROM), or cervical incompetency (CI) [4,5,6,7]. Moreover, several clinical studies have assessed whether antibiotic treatment of abnormal vaginal flora (as in bacterial vaginosis) reduces preterm birth rates in high-risk and even low-risk women: however, antibiotics were found to be ineffective, contrary to many expectations [8].

There have been many investigations of the risk factors for abnormal vaginal flora, but these mainly targeted the non-pregnant population. Previously identified risk factors for bacterial vaginosis among women of reproductive age included low socioeconomic status, black ethnicity, and early sexual experience [9,10]. Use of intrauterine devices, history of past spontaneous abortion, and menstrual cycle over 35 days were also identified as risk factors for bacterial vaginosis in non-pregnant women [11]. With regard to aerobic vaginitis, which has emerged as a relatively new disease entity [12], unmarried status, use of intrauterine devices, long-term use of antibiotics (≥10 days in the past month), and frequent vaginal douching were found to be independent risk factors in women of reproductive age [13]. For genital mycoplasmas including *Ureaplasma urealyticum*, younger age, sexual behaviour, and low socioeconomic status are known risk factors [14,15,16].

Previously, we found that few comprehensive studies regarding maternal risk factors for abnormal vaginal flora during pregnancy have been published and have significant limitations. These studies have been largely focused on bacterial vaginosis rather than aerobic bacterial colonisation or genital mycoplasmas. In addition, the study designs included low-risk pregnant women, making it unclear whether the presence of abnormal vaginal flora adversely influences high-risk pregnancies among women who already had signs and symptoms indicative of spontaneous preterm birth.

With this background, we aimed to identify the maternal risk factors for abnormal vaginal flora, caused by colonisation of aerobic microorganisms or genital mycoplasmas, in pregnant women at high risk of spontaneous preterm birth. We also examined whether abnormal vaginal flora is associated with adverse outcomes in these women.

## 2. Materials and Methods

This retrospective cohort study included consecutive parturient women who were admitted to our institution’s high-risk care unit and underwent vaginal culture at admission between January 2010 and July 2020. The inclusion criteria were patients whose gestational ages were 14 0/7–34 6/7 weeks at admission (*n* = 1381) and the presence of one of the following indications: PTL, pPROM, CI, and short cervical length (CL). The diagnoses were mutually exclusive. Women diagnosed with PTL required admission due to regular uterine contraction and progressive cervical change. Women with pPROM were diagnosed based on gross pooling of amniotic fluid in the vagina and an alkaline vaginal fluid pH determined by a nitrazine test. Women with CI were those with painless cervical dilatation (>1 cm) with visible membranes through the cervical os. Short CL was defined when the cervix is closed on a sterile speculum, and the cervical length is less than 25 mm when measured through transvaginal ultrasound.

In accordance with our institutional protocols, we routinely perform vaginal culture of every patient at high risk of spontaneous preterm birth at admission under these indications. Vaginal sampling for culture was performed by swabbing the posterior fornix using a sterile cotton swab and a sterile non-lubricated speculum. The samples were transported for analysis by Copan Venturi Transystem collection device (Copan innovation, Corona, CA, USA), which is composed of Liquid Stuart transport medium. The Gram-staining and culture were carried out according to the standard laboratory procedures of our institution. The samples were directly inoculated on a blood agar plate, MacConkey agar plate, and Thayer–Martin agar plate and were incubated for up to 72 h at 35 °C in a 5% CO_2_ incubator. Results of vaginal culture were categorised by the types of colonised microorganism; bacteria (Gram-negative and -positive) and genital mycoplasmas (*Mycoplasma hominis* and *Ureaplasma urealyticum*). Abnormal vaginal flora was defined as isolation of any of the above-mentioned microorganisms. Colonisation with either Gram-negative or -positive bacteria was defined as abnormal bacterial colonisation.

Data were collected from the review of the medical records and were compared according to the category of vaginal flora. Maternal baseline characteristics collected included maternal age and pre-pregnancy body mass index (BMI). Parity was reviewed in detail, including the number of past spontaneous and induced abortions. With regard to socioeconomic factors, marital status, smoking history, drinking history after pregnancy, and highest level of education completed were recorded. Marital status was categorised as either single or married/common-law. The highest level of education completed was categorised as lower than high-school graduate, high-school graduate, some college/college graduate, or higher than college graduate. Clinical characteristics collected included gestational age at admission, diagnosis at admission (PTL, pPROM, CI, or short CL), multifetal pregnancy, and infertility treatment (intrauterine insemination [IUI] or in vitro fertilisation [IVF]) for the index pregnancy.

We collected the pregnancy outcome of every foetus of the study population that was alive at the time of admission (*n* = 1648). Pregnancy outcome was categorised as abortion, stillbirth/foetal death in uterus (FDIU), or livebirth. Abortion included all foetal loss in utero occurring before 20 weeks of gestation, while stillbirth/FDIU included foetal loss occurring beyond 20 weeks of gestation with 5 min Apgar scores of 0. Delivery outcomes included gestational age at delivery, mode of delivery, and diagnosis of histological chorioamnionitis and were analysed for the 1381 mothers. Neonatal outcome was assessed for the 1519 liveborn neonates of the study population. The variables included sex, birth weight, Apgar score, neonatal intensive care unit (NICU) admission, diagnosis of neonatal sepsis, and neonatal mortality. Neonates with birth weights below the 10th percentile for gestational age were defined as small for gestational age (SGA). Neonatal sepsis was specified as early onset neonatal sepsis (EONS) when diagnosed within the first week of life, the subsequent diagnosis was classified as late-onset neonatal sepsis (LONS). Suspected EONS was diagnosed clinically by individual paediatricians, whereas proven EONS was diagnosed upon the isolation of microorganisms from blood or cerebrospinal fluid of the neonates.

All statistical analyses were completed using SPSS^®^ ver. 27.0 (IBM Corp., Armonk, NY, USA) and the results were considered statistically significant when the *p* value was <0.05. Categorical variables were analysed using Pearson’s chi-square test or Fisher’s exact test, and continuous data were tested using the Mann–Whitney U test when appropriate. Linear-by-linear association was used to evaluate the statistical significance of trends in the number of past abortions and level of education, according to abnormal vaginal colonisation. To determine whether abnormal vaginal colonisation is a meaningful factor even after adjusting factors which are generally known to directly affect neonatal outcomes, multivariate analysis was conducted using a logistic regression model. We selected gestational age at delivery and history of preterm delivery as the most influential factors and used them for analysis. Moreover, treatment for vaginal colonisation, namely the use of antepartum antibiotics, would affect the neonatal outcomes, so we also adjusted it for analysis. The use of antepartum antibiotics is defined as use of antibiotics of any kind for 2 or more days after admission, since using antibiotics one day or less is less likely to affect neonatal outcomes. The history of antibiotics was unclear in 56 patients (65 foetuses) and were excluded from the analysis.

## 3. Results

A total of 1381 parturient women were included in this study, and abnormal vaginal bacterial flora was identified in 283 cases (20.5%). A total of 178 (12.9%) and 124 (9.0%) cases had at least one of Gram negative and positive microbes isolated, respectively. The most frequently isolated Gram-negative bacterium was *E. coli* (*n* = 122, 8.8%), followed by *Klebsiella* (*n* = 29, 2.1%), *Enterobacter* (*n* = 14, 1.0%), *Citrobacter* (*n* = 10, 0.7%), *Pseudomonas* (*n* = 6, 0.4%), and *Proteus* (*n* = 6, 0.4%). Gram-positive bacterial colonisation was mostly due to group B *Streptococcus* (GBS) (*n* = 61, 4.4%), followed by coagulase-negative *Staphylococcus* (*n* = 18, 1.3%), and *S. anginosus* (*n* = 4, 0.3%). Results of mycoplasma culture were available for 1309 patients. Among these, the colonisation by genital mycoplasmas occurred in 43.6% (*n* = 571), comprising 4.4% (*n* = 58) *M. hominis* and 42.4% (*n* = 555) *Ureaplasma* species.

Table 1 presents the maternal baseline characteristics, socioeconomic factors, and clinical characteristics in the Gram-negative or -positive bacterial colonisation groups compared to the group with no bacterial colonisation. Vaginal Gram-negative bacterial colonisation was associated with advanced maternal age, increased pre-pregnancy BMI, greater numbers of past spontaneous abortions, and pregnancy through IVF as compared with the group with no bacterial colonisation. In contrast, we identified no baseline or socioeconomic risk factors for Gram-positive bacterial colonisation. Both Gram-negative and -positive bacterial colonisation was associated with an earlier gestational age at admission as compared with no bacterial colonisation. With regard to the diagnosis of admission, mothers with Gram-negative bacteria colonisation were more likely to be admitted for CI (15.1% vs. 33.7%, *p* < 0.001) and less likely to be admitted for PTL (36.4% vs. 28.7%, *p* = 0.044), but there was no difference in the diagnosis of admission according to the presence of Gram-positive bacteria colonisation.

Table 2 summarises the maternal baseline characteristics, socioeconomic factors, and clinical characteristics in the group with genital mycoplasmal colonisation and the group with no genital mycoplasma colonisation. Most of the factors were not significantly different between the two groups. Of note, a greater number of past induced abortions, lower level of education completed, and lower rates of multifetal pregnancy and IVF were found in the genital mycoplasmal colonisation group. The genital mycoplasmal colonisation group was more likely to be admitted for pPROM compared to those without colonisation (42.4% vs. 35.6%, *p* = 0.013).

Table 3 presents the pregnancy, delivery, and neonatal outcomes in the Gram-negative or -positive bacterial colonisation groups and the group with no bacterial colonisation. The Gram-negative bacterial colonisation group had a greater number of stillbirths/FDIU (6.5% vs. 3.1%, *p* = 0.014), and delivered babies of an earlier gestational age (27 4/7 weeks [16 1/7 to 37 6/7] vs. 30 4/7 weeks [16 0/7 to 38 4/7], *p* < 0.001) compared with the group with no bacterial colonisation. The Gram-positive bacterial colonisation group showed a similar tendency: higher rate of stillbirth/FDIU (8.3% vs. 3.1%, *p* = 0.002) and an earlier gestational age at delivery (27 6/7 weeks [16 0/7 to 37 4/7] vs. 30 4/7 weeks [16 0/7 to 38 4/7], *p* = 0.002). Association with histological chorioamnionitis was only evident in the Gram-negative colonisation group (57.4% vs. 43.6%, *p* < 0.001). In addition, associations with neonatal morbidity and mortality were found only in the Gram-negative colonisation group: specifically, the group’s infants had a higher risk of NICU admission (97.4% vs. 89.4%, *p* < 0.001) and proven EONS (3.1% vs. 1.1%, *p* = 0.042). There was no significant difference in neonatal outcomes between the Gram-positive and no bacterial colonisation groups. After adjustment of gestational age at delivery, history of preterm birth, and use of antibiotics two or more days after admission, only risk of NICU admission remained significantly higher in the Gram-negative colonisation group (OR [95% CI]: 4.179 [1.471–11.873], *p* = 0.007), and other pregnancy, delivery, and neonatal outcomes did not. After adjustment, there were still no factors related to the Gram-positive colonisation group.

Table 4 presents the pregnancy, delivery, and neonatal outcomes in the genital mycoplasmal colonisation group compared and the group with no genital mycoplasma colonisation. Genital mycoplasmal colonisation was associated with histological chorioamnionitis (50.4% vs. 42.9%, *p* = 0.008), but was not associated with any of the other adverse pregnancy outcomes except for a greater percentage of 5 min Apgar score < 7. The association with histological chorioamnionitis was still statistically significant after adjustment (OR [95% CI]: 1.281 [1.005–1.633], *p* = 0.045), while other outcome factors was not.

## 4. Discussion

Abnormal bacterial colonisation by aerobic microorganisms during pregnancy has been reported previously. According to a Belgian study that included non-symptomatic pregnant women, the prevalence of abnormal vaginal flora during the first trimester was 9.35% [17]. In studies including high-risk pregnant women with PTL or pPROM, the rate of aerobic vaginitis ranged from 11.3% to 18% [18,19]. In our study population including high-risk pregnant women (PTL, pPROM, CI, and short CL), aerobic bacterial colonisation was detected in 20.5% of the women, and the most frequently observed microbes were *E. coli* (8.8%) and GBS (4.4%). These two microbes were reported to be among the most common causative organisms of aerobic vaginitis in previous studies [12,20].

In this study, we found that risk factors for aerobic bacterial colonisation during pregnancy manifested distinct patterns from those of bacterial vaginosis. In general, maternal baseline risk factors for bacterial vaginosis in pregnancy have been reported to be smoking at conception, high alcohol consumption, being single, and lower level of education [21,22], none of which were found to be significant risk factors for either Gram-negative or -positive bacterial colonisation in our analyses. In contrast, maternal age, which was not previously reported to be associated with bacterial vaginosis [21,22], was found to be a significant risk factor for Gram-negative colonisation in this study. In addition, the history of past spontaneous abortion, which was the risk factor of bacterial vaginosis in reproductive women [11], was common. Specifically, we found a linear association between the number of past spontaneous abortions and Gram-negative bacteria colonisation. Our finding that IVF pregnancies are risk factors for abnormal vaginal colonisation is an extension of the results of our previous study, which showed that the prevalence of vaginal colonisation by Gram-negative bacteria, especially *Escherichia coli*, was significantly greater in pregnancies resulting from infertility treatment after adjustment for confounding variables [23].

It is noteworthy that neonates from both the Gram-negative and -positive colonisation groups had earlier gestational age at delivery in our study, but only the Gram-negative group had significantly higher rate of histological chorioamnionitis. This result is partly consistent with the results of a previous study showing that Gram-negative rods (but not Gram-positive cocci) were the only chorioamniotic isolate to be significantly correlated with histological chorioamnionitis [24]. In addition, our study showed that only the Gram-negative colonisation group was associated with adverse neonatal outcomes, such as NICU admission and proven EONS. The risk of NICU admission was significantly higher in the Gram-negative group even after the adjustment of other risk factors. Bacteria from the vaginal flora can migrate directly to foetal membranes, causing inflammation and consequent neonatal morbidity and mortality [25]. Our results suggest that Gram-negative bacteria are better able to produce ascending infection than Gram-positive bacteria and produce greater impact on neonatal outcomes. In fact, a recent in vitro study demonstrated that lipopolysaccharides in Gram negative bacteria induce time-dependent and cell-type-specific pro-inflammatory cytokines, which contribute to propagation though decidua, chorion, and amnion [26].

Genital mycoplasmas are sometimes considered part of the normal vaginal microbiota but can also be associated with adverse pregnancy outcomes, especially where bacterial loads are increased or in combination with other abnormal vaginal flora such as bacterial vaginosis in preterm gestation [15,27]. In general, the risk factors for mycoplasmal infection in non-pregnant women are younger age, risky sexual behaviour, and a lower socioeconomic status [15]. However, it is not clear that these risk factors also applied during pregnancy when the vaginal microbiome changes physiologically. In our study, we confirmed that an increased number of artificial abortions and lower education level tended to be associated with genital mycoplasmal colonisation in high-risk pregnant women. Unexpectedly, we found that genital mycoplasmas were associated with reduced multifetal pregnancies and infertility treatment. We presume that infertility treatment protocols including doxycycline administration before hysterosalpingography may reduce colonisation by genital mycoplasmas, as previously noted [23,28].

Unlike the robust intra-amniotic infection or inflammation caused by *Ureaplasma* in amniotic fluid [29], the impact of maternal vaginal colonisation by genital mycoplasmas on pregnancy outcomes is less overt [27,30,31]. It was suggested in a meta-analysis that genital mycoplasma presence in the vagina is insufficient to promote preterm birth [32]. For example, for *Ureaplasma urealyticum,* the odds ratio for preterm labour was 1.94 [0.77–4.90] for maternal samples, but 2.68 [1.18–6.09] for foetal samples [32]. In our study including high-risk pregnant women, associations between vaginal genital mycoplasmas and adverse pregnancy outcomes were not demonstrated except for histological chorioamnionitis. Such associations between vaginal *Ureaplasma* species and histologic chorioamnionitis were previously demonstrated in preterm births at less than 37 weeks [33] and in very preterm infants with pPROM [34]. According to a Japanese study, isolation of *Ureaplasma* spp. in the placenta from preterm births was associated with the severity of histologic chorioamnionitis [35]. In addition, the urease structural gene and protein of *Ureaplasma* spp. were detected in the amnion, indicating direct infection of *Ureaplasma* spp. [35].

The clinical significance of antibiotic treatment for adverse vaginal colonisation in pregnant women is uncertain. However, there are studies that suggest antibiotic therapy may be more helpful earlier in pregnancy, since the irreversible negative effects of abnormal vaginal microorganism are assumed to happen during that period [36,37]. Our finding that patients in both the Gram-negative and -positive bacterial colonisation groups were admitted for symptoms of preterm birth at earlier gestational age is also consistent with this assertion.

There are some limitations of our study. Most of the former studies assessing abnormal vaginal flora classified the concept as bacterial vaginosis, aerobic vaginitis, mixed abnormal flora, or intermediate flora. For this classification, wet-mount microscopy, whiff test, and pH measurement of the vaginal discharge are required in addition to the identification of bacteria [7,38,39]. However, due to limited resources to perform such tests at our institution, only the identification of the microorganism by culture was completed; this precluded us from analysing the maternal inflammatory response to the colonised bacteria. Among genital mycoplasmas, only *Mycoplasma hominis* and *Ureaplasma urealyticum* were checked, so it could not be analyzed for other genital mycoplasmas, including *Mycoplasma genitalium*, infection of which is known to be associated with preterm birth [40]. Additionally, there is a possibility that treatments for preterm birth or other medical conditions before admission might have altered the maternal vaginal flora [41]. The representativeness of vaginal culture at admission might be a matter of debate.

Nonetheless, we consider our study to be clinically meaningful since there have been relatively few studies addressing the risk factors and effects of abnormal vaginal flora on parturient women who have already shown manifestations of preterm delivery. In addition, having gathered data from a relatively large number of consecutive patients (*n* = 1381) added strength to the results. While current guidelines indicate screening for bacterial vaginosis in low-risk pregnant women is not beneficial, routine screening and treatment of abnormal vaginal flora in high-risk pregnant women is still a matter of debate [42]. A prospective study of antibiotic therapy focusing on Gram-negative microbes should also be considered within the larger goal of reducing preterm births. With the prospective study, it may be justified to conduct vaginal cultures to detect Gram-negative colonisation, at least among women with high-risk pregnancies.

## Figures and Tables

**Table 1 jcm-12-00040-t001:** Maternal baseline characteristics, socioeconomic factors, and clinical characteristics in the groups with vaginal colonisation by Gram-negative or -positive bacteria compared with the group with no bacterial colonisation.

	Maternal Vaginal Colonisation at Admission *				
	No Bacterial Colonisation (*n* = 1098)	Gram (−) Bacterial Colonisation (*n* = 178)	*p* Value	Gram (+) Bacterial Colonisation (*n* = 124)	*p* Value
** *Maternal baseline characteristics* **					
**Age [years]**	33 (18~45)	34 (20~42)	0.007	33 (19~43)	0.057
**Pre-pregnancy BMI [kg/m^2^] ^†^**	20.86 (14.04~43.41)	21.45 (15.55~47.34)	0.006	21.23 (14.90~35.52)	0.159
** *Parity* **					
**Nulliparity**	60.4% (663/1098)	65.2% (116/178)	0.224	59.7% (74/124)	0.879
**History of preterm delivery ^‡^**	25.5% (111/435)	29.0% (18/62)	0.555	24.0% (12/50)	0.815
**Past spontaneous abortion [count]**			0.010		0.184
none	77.1% (847/1098)	68.5% (122/178)		82.3% (102/124)	
1	15.1% (166/1098)	19.7% (35/178)		13.7% (17/124)	
2	5.5% (60/1098)	7.3% (13/178)		1.6% (2/124)	
≥3	2.3% (25/1098)	4.5% (8/178)		2.4% (3/124)	
**Past induced abortion [count]**			0.577		0.192
none	92.2% (1012/1098)	89.9% (160/178)		87.9% (109/124)	
1	5.4% (59/1098)	7.3% (13/178)		8.1% (10/124)	
2	1.5% (17/1098)	2.2% (4/178)		3.2% (4/124)	
≥3	0.9% (10/1098)	0.6% (1/178)		0.8% (1/124)	
** *Socioeconomic factors* **					
**Marital status**			0.073		1.000
Single	0.7% (8/1098)	2.2% (4/178)		0.8% (1/124)	
Married/Common-law	99.3% (1090/1098)	97.8% (174/178)		99.2% (123/124)	
**Smoking history**					
Any before or after pregnancy ^†^	2.0% (15/742)	1.3% (2/155)	0.751	0.0% (0/109)	0.239
**Drinking history**			1.000		1.000
Any after pregnancy	0.5% (5/1098)	0.0% (0/178)		0.0% (0/124)	
**Highest level of education ^†^**			0.167		0.479
Lower than high-school graduate	1.3% (13/1014)	2.9% (4/138)		0.0% (0/87)	
High-school graduate	14.9% (151/1014)	13.8% (19/138)		12.6% (11/87)	
Some college/college graduate	72.4% (734/1014)	76.8% (106/138)		75.9% (66/87)	
Higher than college graduate	11.4% (116/1014)	6.5% (9/138)		11.5% (10/87)	
** *Clinical characteristics of index pregnancy* **					
**Gestational age at admission [weeks]**	28 6/7 (15 6/7 to 34 6/7)	25 4/7 (14 4/7 to 34 6/7)	<0.001	26 2.5/7 (15 6/7 to 34 6/7)	0.005
**Diagnosis at admission**					
PTL	36.4% (400/1098)	28.7% (51/178)	0.044	29.0% (36/124)	0.103
pPROM	37.9% (416/1098)	31.5% (56/178)	0.099	40.3% (50/124)	0.597
CI	15.1% (166/1098)	33.7% (60/178)	<0.001	21.0% (26/124)	0.090
short CL	4.1% (45/1098)	2.2% (4/178)	0.233	4.8% (6/124)	0.696
**Multifetal pregnancy ^§^**	20.5% (225/1098)	22.5% (40/178)	0.546	18.5% (23/124)	0.610
Twin ^§^	19.0% (205/1078)	20.2% (35/173)	0.706	17.2% (21/122)	0.629
Triplet ^§^	2.2% (20/893)	3.5% (5/143)	0.374	1.9% (2/103)	1.000
**Infertility treatment** ** ^¶^ **	18.7% (205/1098)	31.5% (56/178)	<0.001	19.4% (24/124)	0.853
IUI **^¶^**	2.2% (20/913)	3.2% (4/126)	0.521	4.8% (5/105)	0.169
IVF **^¶^**	17.2% (185/1078)	29.9% (52/174)	<0.001	16.0% (19/119)	0.742

BMI: body mass index; PTL: preterm labour; pPROM: preterm premature rupture of membranes; CI: cervical incompetence; short CL: short cervical length; IUI: intrauterine insemination; IVF: in vitro fertilisation. Data are presented as the median (range) or percentage (number). * 1381 high-risk parturient women who underwent vaginal culture at admission were included in analysis. ^†^ Data regarding pre-pregnancy BMI, history of smoking before pregnancy, and final education were not available for some patients (Data existing in *n* = 1079, 987, 1231 each). ^‡^ Analysed for multiparous women only. ^§^ Compared with the singleton pregnancy group. **^¶^** Compared to the group that did not receive any infertility treatment.

**Table 2 jcm-12-00040-t002:** Maternal baseline characteristics, socioeconomic factors, and clinical characteristics in the group with vaginal colonisation by genital mycoplasmas compared with the group with no mycoplasmal colonisation.

	Maternal Vaginal Colonisation at Admission *		*p* Value
	No Genital Mycoplasmal Colonisation (*n* = 738)	Genital Mycoplasmal Colonisation (*n* = 571)	
** *Maternal baseline characteristics* **			
**Age [years]**	33 (18~45)	33 (18~45)	0.659
**Pre-pregnancy BMI [kg/m^2^] ^†^**	20.93 (14.90~43.41)	21.23 (14.04~47.34)	0.052
** *Parity* **			
**Nulliparity**	61.2% (452/738)	60.6% (346/571)	0.811
**History of preterm delivery ^‡^**	23.4% (67/286)	29.8% (67/225)	0.105
**Past spontaneous abortion [count]**			0.635
none	75.7% (559/738)	76.5% (437/571)	
1	16.0% (118/738)	15.4% (88/571)	
2	5.3% (39/738)	6.0% (34/571)	
≥3	3.0% (22/738)	2.1% (12/571)	
**Past induced abortion [count]**			0.021
none	92.8% (685/738)	89.7% (512/571)	
1	5.3% (39/738)	6.3% (36/571)	
2	1.2% (9/738)	2.8% (16/571)	
≥3	0.7% (5/738)	1.2% (7/571)	
** *Socioeconomic factors* **			
**Marital status**			0.455
Single	0.8% (6/738)	1.2% (7/571)	
Married/Common-law	99.2% (732/738)	98.8% (564/571)	
**Smoking history**			
Any before or after pregnancy ^†^	0.9% (4/457)	2.5% (12/477)	0.053
**Drinking history**			1.000
Any after pregnancy	0.4% (3/738)	0.4% (2/571)	
**Highest level of education ^†^**			0.022
Lower than high-school graduate	0.9% (6/675)	2.0% (10/497)	
High-school graduate	14.2% (96/675)	15.3% (76/497)	
Some college/college graduate	72.1% (487/675)	74.2% (369/497)	
Higher than college graduate	12.7% (86/675)	8.5% (42/497)	
** *Clinical characteristics of index pregnancy* **			
**Gestational age at admission [weeks]**	28 4/7 (14 4/7 to 34 6/7)	28 0/7 (15 6/7 to 34 6/7)	0.922
**Diagnosis at admission**			
PTL	36.7% (271/738)	32.9% (188/571)	0.153
pPROM	35.6% (263/738)	42.4% (242/571)	0.013
CI	18.6% (137/738)	17.0% (97/571)	0.461
short CL	3.8% (28/738)	4.2% (24/571)	0.707
**Multifetal pregnancy ^§^**	24.7% (182/738)	15.2% (87/571)	<0.001
Twin ^§^	23.0% (166/722)	14.0% (79/563)	<0.001
Triplet ^§^	2.8% (16/572)	1.6% (8/492)	0.200
**Infertility treatment** ** ^¶^ **	24.1% (178/738)	15.4% (88/571)	<0.001
IUI ^¶^	1.8% (10/570)	3.2% (16/499)	0.124
IVF ^¶^	23.1% (168/728)	13.0% (72/555)	<0.001

BMI: body mass index; PTL: preterm labour; pPROM: preterm premature rupture of membranes; CI: cervical incompetence; short CL: short cervical length; IUI: intrauterine insemination; IVF: in vitro fertilisation. Data are presented as median (range) or percentage (number). * 1309 high-risk parturient women who underwent vaginal culture at admission were included in the analyses. ^†^ Data for pre-pregnancy BMI, history of smoking before pregnancy, education data were not available for some patients. (Data existing in *n* = 1017, 934, 1172 each). ^‡^ Analysed for multiparous women only. ^§^ Compared with the singleton pregnancy group. **^¶^** Compared with the group that did not receive any infertility treatment.

**Table 3 jcm-12-00040-t003:** Pregnancy, delivery, and neonatal outcomes in groups with vaginal colonisation by Gram-negative or -positive bacteria compared with the group with no bacterial colonisation.

	Maternal Vaginal Colonisation at Admission				
	No Bacterial Colonisation	Gram (−) Bacterial Colonisation	*p* Value	Gram (+) Bacterial Colonisation	*p* Value
** *Pregnancy outcome (n = 1648) ** **	n = 1311	n = 216		n = 145	
Abortion	3.5% (46/1311)	5.1% (11/216)	0.255	6.2% (9/145)	0.106
Stillbirth/FDIU	3.1% (41/1311)	6.5% (14/216)	0.014	8.3% (12/145)	0.002
Livebirth	93.4% (1224/1311)	88.4% (191/216)	0.010	85.5% (124/145)	<0.001
** *Delivery outcome (n = 1381)* ^†^ **	n = 1098	n = 178		n = 124	
**Gestational age at delivery [weeks]**	30 4/7 (16 0/7 to 38 4/7)	27 4/7 (16 1/7 to 37 6/7)	<0.001	27 6/7 (16 0/7 to 37 4/7)	0.002
**Caesarean section**	53.9% (592/1098)	56.2% (100/178)	0.574	44.4% (55/124)	0.043
**Histological chorioamnionitis ^§^**	43.6% (470/1077)	57.4% (101/176)	<0.001	50.8% (61/120)	0.132
** *Neonatal outcome (n = 1519)* ^‡^ **	n = 1224	n = 191		n = 124	
**Sex [male]**	55.2% (676/1224)	53.4% (102/191)	0.637	60.5% (75/124)	0.262
**Birth weight [g]**	1570 (84~4110)	1140 (400~3130)	<0.001	1300 (380~2870)	0.068
**SGA**	8.3% (102/1224)	11.0% (21/191)	0.225	4.8% (6/124)	0.172
**1 min Apgar score <4**	10.0% (122/1224)	12.6% (24/191)	0.272	9.7% (12/124)	0.918
**5 min Apgar score <7**	11.8% (144/1224)	12.6% (24/191)	0.750	14.5% (18/124)	0.369
**NICU admission**	89.4% (1094/1224)	97.4% (186/191)	<0.001	90.3% (112/124)	0.744
**Neonatal sepsis**	7.0% (86/1224)	9.9% (19/191)	0.152	10.5% (13/124)	0.160
LONS	4.2% (52/1224)	4.7% (9/191)	0.769	5.6% (7/124)	0.469
EONS	2.8% (34/1224)	5.2% (10/191)	0.069	4.8% (6/124)	0.258
proven	1.1% (14/1224)	3.1% (6/191)	0.042	3.2% (4/124)	0.076
suspected	1.6% (20/1224)	2.1% (4/191)	0.554	1.6% (2/124)	1.000
**Neonatal mortality**	6.2% (76/1224)	9.9% (19/191)	0.055	7.3% (9/124)	0.647

FDIU: foetal death in uterus; SGA: small for gestational age; NICU: neonatal intensive care unit; LONS: late-onset neonatal sepsis; EONS: early onset neonatal sepsis. Data are presented as the median (range) or percentage (number). * 1648 foetuses whose mothers were admitted to the high-risk pregnancy centre were included for analysis. Vanishing or demised twins and foetuses that underwent selective foetal reduction were excluded from the study population. ^†^ 1381 high-risk parturient women who underwent vaginal culture at admission and delivered at the centre were included in the analyses. ^‡^ 1519 liveborn neonates whose mothers were admitted to the high-risk pregnancy centre were included in the analyses. ^§^ Histological examination results were not available for 27 patients.

**Table 4 jcm-12-00040-t004:** Pregnancy, delivery, and neonatal outcomes in the group with vaginal colonisation by genital mycoplasmas compared with the group with no mycoplasmal colonisation.

	Maternal Vaginal Colonisation at Admission		*p* Value
	No Genital Mycoplasmal Colonisation	Genital Mycoplasmal Colonisation	
** *Pregnancy outcome (n = 1564) ** **	n = 911	n = 653	
Abortion ^†^	4.1% (37/911)	3.4% (22/653)	0.478
Stillbirth/FDIU ^†^	3.3% (30/911)	4.4% (29/653)	0.240
Livebirth ^†^	92.6% (844/911)	92.2% (602/653)	0.737
** *Delivery outcome (n = 1309)* ^†^ **	n = 738	n = 571	
**Gestational age at delivery [weeks]**	30 1/7 (16 0/7 to 38 4/7)	30 2/7 (16 0/7 to 37 6/7)	0.745
**Caesarean section**	55.8% (412/738)	50.3% (287/571)	0.045
**Histological chorioamnionitis ^§^**	42.9% (308/718)	50.4% (284/564)	0.008
** *Neonatal outcome (n = 1446)* ^‡^ **	n = 844	n = 602	
**Sex [male]**	56.3% (475/844)	54.0% (325/602)	0.387
**Birth weight [g]**	1490 (84~3230)	1565 (300~4110)	0.097
**SGA**	8.4% (71/844)	7.8% (47/602)	0.679
**1 min Apgar score <4**	9.2% (78/844)	11.8% (71/602)	0.116
**5 min Apgar score <7**	10.8% (91/844)	14.3% (86/602)	0.045
**NICU admission**	90.4% (763/844)	91.0% (548/602)	0.686
**Neonatal sepsis**	6.6% (56/844)	8.1% (49/602)	0.277
LONS	3.9% (33/844)	4.8% (29/602)	0.401
EONS	2.7% (23/844)	3.3% (20/602)	0.510
proven	1.3% (11/844)	1.5% (9/602)	0.758
suspected	1.4% (12/844)	1.8% (11/602)	0.544
**Neonatal mortality**	6.0% (51/844)	7.1% (43/602)	0.403

FDIU: foetal death in uterus; SGA: small for gestational age; NICU: neonatal intensive care unit; LONS: late-onset neonatal sepsis; EONS: early onset neonatal sepsis. Data are presented as median (range) or percentage (number). * 1564 foetuses whose mothers were admitted to the high-risk pregnancy centre were included in the analyses. Vanishing or demised twins and foetuses that underwent selective foetal reduction were excluded from the study population. ^†^ 1309 high-risk parturient women who underwent vaginal culture at admission and delivered at the centre were included in the analyses. ^‡^ 1446 liveborn neonates whose mothers were admitted to the high-risk pregnancy centre were included in the analyses. ^§^ Results of histological examination were not available in 27 patients.

## Data Availability

The datasets analyzed during the current study are available from the corresponding author on reasonable request, under approval of Institutional Review Board of our institution.

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
