# Peer review of "Maternal Baseline Risk Factors for Abnormal Vaginal Colonisation among High-Risk Pregnant Women and the Association with Adverse Pregnancy Outcomes: A Retrospective Cohort Study"

_jcm, 2022, doi:10.3390/jcm12010040_

Round 1

Reviewer 2 Report

Major issues

lines 77-78. It is not clear when the vaginal swabs were collected. In the Methods you state that they were collected "at admission". However in the Discussion you state: "Also there is a possibility that treatments for preterm birth or other medical conditions before and after admission might have altered the maternal vaginal flora". These two statements are inconsistent with regard to treatments after admission altering the flora which was sampled at admission.

Furthermore you do not provide any details of the microbiological procedures followed, eg agar used, duration of incubation, etc. Nor do you state whether the bacteria isolated were present in pure growth or mixed growth.

lines 121-127. The number of patients from whom Gram positive or Gram negative bacteria were isolated is confusing . You state that: "Gram negative and positive microbes were isolated in  178 and 124 cases respectively." However, from the data provided in the following sentence, the sum of the Gram negative cases is 187, and the sum of the Gram positive cases is 83. The discrepancies need to be corrected or else explained.

Lines 254-6. I do not believe that an in vitro study can show that "lipopolysaccharides in Gram negative bacteria contribute to bacterial propagation in the uterus in ascending infection". I think a more cautious wording may be more appropriate.

I think that you need to be more clear about whether vaginal colonisation itself, or intrauterine infection arising from vaginal colonisation is the significant factor in adverse pregnancy outcomes. Lines 272-3 imply that colonisation itself may have an adverse effect, but that seems unlikely.

Further confusion arises in lines 283-5 where you suggest that Ureaplasma might cause"colonisation" of the amnion. I think the more appropriate term would be "infection".

Your last two sentences in the Discussion seem to be back to front. Unless a prospective study demonstrates that antibiotic therapy of vaginal colonisation by Gram negative bacteria does improve pregnancy outcomes there would not seem to be a strong justification for screening to detect such colonisation.

Minor issues

lines 39-40. this sentence needs improving eg by adding "whether" and deleting "the use of"after "have assessed"

line 70. you have not provided the full term before the first use of the abbreviation "CL".

In Table 1. there appears to be an extra p value for "Smoking history" analyses for both the Gram +ve and the Gram -ve columns.

Lines 234-6. This sentence has an "as" and an "in" which are not required.

lines 263-4. The phrase: "when the change of vaginal microbiome is occurred either in physiological aspect or in association with preterm birth." is very unclear. It requires rewriting.
